# Artificial Neural Networks for Predicting Plastic Anisotropy of Sheet Metals Based on Indentation Test

**DOI:** 10.3390/ma15051714

**Published:** 2022-02-24

**Authors:** Jiaping Xia, Chanhee Won, Hyunggyu Kim, Wonjoo Lee, Jonghun Yoon

**Affiliations:** 1Department of Mechanical Design Engineering, Hanyang University, 222, Wangsimni-ro, Seongdong-gu, Seoul 04763, Korea; xiajiaping666@gmail.com (J.X.); gudrb8535@gmail.com (H.K.); wj6478@gmail.com (W.L.); 2Department of Mechanical Engineering, BK21 FOUR ERICA-ACE Center, Hanyang University, 55 Hanyangdaehak-ro, Sangnok-gu, Ansan 15588, Korea; 3Digital Transformation R&D Department, Korea Institute of Industrial Technology, 143, Hanggaul-ro, Sangrok-gu, Ansan 15588, Korea; chan2@kitech.re.kr; 4AIDICOME Inc., 55, Hanyangdaehak-ro, Sangnok-gu, Ansan 15588, Korea

**Keywords:** plastic anisotropy, stress–strain flow curve, Lankford coefficient, spherical indentation, artificial neural network, residual indentation mark, machine vision

## Abstract

This paper mainly proposes two kinds of artificial neural network (ANN) models for predicting the plastic anisotropy properties of sheet metal using spherical indentation test, which minimizes measurement time, costs, and simplifies the process of obtaining the anisotropy properties than the conventional tensile test. The proposed ANN models for predicting anisotropic properties can replace the traditional complex dimensionless analysis. Moreover, this paper is not limited to the prediction of yield strength anisotropy but also further accurately predicts the Lankford coefficient in different orientations. We newly construct an FE spherical indentation model, which is suitable for sheet metal in consideration of actual compliance. To obtain a large dataset for training the ANN, the constructed FE model is utilized to simulate pure and alloyed engineering metals with one thousand elastoplastic parameter conditions. We suggest the specific variables of the residual indentation mark as input parameters, also with the indentation load–depth curve. The profile of the residual indentation, including the height and length in different orientations, are used to analyze the anisotropic properties of the material. Experimental validations have been conducted with three different sheet alloys, TRIP1180 steel, zinc alloy, and aluminum alloy 6063-T6, comparing the proposed ANN model and the uniaxial tensile test. In addition, machine vision was used to efficiently analyze the residual indentation marks and automatically measure the indentation profiles in different orientations. The proposed ANN model exhibits remarkable performance in the prediction of the flow curves and Lankford coefficient of different orientations.

## 1. Introduction

Sheet metal forming is widely used in automotive, aerospace, shipbuilding, chemical, nuclear, and other manufacturing processes. During the production of the sheet, changes in the microstructure of the metal cause significant differences in the strength characteristics of the material, producing the anisotropy of the sheet metal. The plastic anisotropy of pure and synthetic materials has a non-negligible effect on their formation performance [1,2,3], affecting the formability of blanks, which is in some cases very serious, including wrinkles and thinning failures. The study of plastic anisotropy has a great impact on the processing and manufacturing process of materials, such as punching, and it is also of great significance to the technology of Wire Arc Additive Manufacturing (WAAM), widely using aluminum alloys [4]. In general, during the sheet metal forming, the uniaxial tensile test is used to measure the material anisotropy to obtain the stress–strain curve in different orientations, including Young’s modulus, yield strength, hardness exponent, and Lankford coefficient. However, the uniaxial tensile test requires conducting several experiments with the specimens in different orientations, which consumes time and materials. Moreover, the traditional tensile test has certain limitations on the size, shape, and measurement position of the material specimens, and it is challenging to measure the formed sheet. For these reasons, the use of indentation tests to evaluate material properties has emerged recently.

The indentation test is a non-destructive, easier, and faster experimental method that can confirm various mechanical properties of the test material with almost no location, shape, or size restrictions. A reverse analysis method using an indentation test to infer the test material’s elastoplastic properties easily has been proposed and established in the past ten years, the “Oliver–Pharr method” [5]. In addition, the use of indentation tests to estimate the properties of materials, such as plastic flow curve [6,7] and residual stress [8,9,10], is also in progress, while the research on the anisotropy of materials [11,12,13,14] has also become a challenge.

Most research on using indentation testing to extract anisotropy in materials uses inverse dimensional analysis using finite element (FE) simulations with a large number of iterations. Nakamura et al. [11] used the load–depth curves obtained by performing multiple indentation tests with two differently profiled indenter heads, spherical and Berkovich, for extracting the anisotropy of thermally sprayed coatings. However, the method performs multiple experiments instead of a single experiment, which increases the complexity of the estimation. Yoneda et al. [12] showed that the permanent residual indentation impression exhibited an anisotropic shape by conducting an FE analysis and developing a more straightforward method to determine the yield strength, work-hardening exponent, and yield strength ratio. Despite this research only using a single spherical indentation test to estimate, this reverse analysis method has not been verified by actual experiments. Based on the indentation load–depth curve and the residual indentation pile-up height characteristics, Yonezu et al. [13] established a dimensionless function for the anisotropic properties of a material by using a dimensionless analysis to realize the reverse analysis of the anisotropic properties of the material. However, only the anisotropy of the yield strength was studied, and the anisotropy of the strain aspect was not studied. Wang et al. [14] determined an explicit equation relating the plastic anisotropy to the shape characteristic parameters of the bottom of the residual deformation profile to uniquely determine the plastic anisotropy of the material in the section indentation. Although this method can be used when the load–depth curve is not available, this method is as laborious as a traditional tensile test. Wu et al. [15] did not need to use the load–depth curve in the spherical indentation test. Instead, they used only proper orthogonal decomposition to connect the residual indentation mark characteristics with the constitutive parameters of the material: pile-up height and weighting of the average and differences in the residual indentation marks. However, there are unstable factors in the measurement of residual indentation imprinting, and deviation of prediction is more likely to occur after weighting.

The published research demonstrates that the response of the spherical indentation test can fully reflect the anisotropy properties of the materials. The previous methods for extracting anisotropy properties are based on dimensional analyses that correlate the indentation responses with the constitutive parameters of the anisotropic material, focusing on the yield strength ratio of the material in the longitudinal and transverse directions. However, all orientations of the material plane, with the thickness direction, also show the anisotropic properties. Under this consideration, it is essential to find an effective prediction method to predict the anisotropy in different orientations, not only the yield strength ratio but also the Lankford coefficient.

However, in this process, the complex nonlinear relationship between the response of the indentation test and the anisotropic property of the material is a significant obstacle to subsequent prediction. The recent rapid development of data networks, such as machine learning (ML) and machine vision, provides approaches beyond traditional computing to solve these engineering problems. Previous research used ML methods to extract material properties through the indentation test. Huber et al. [16] used a neural network to determine the fixed Poisson’s ratio of material by characterizing the load–depth response in the spherical indentation test as pointwise calculations. Tho et al. [17] constructed an artificial neural network (ANN) with a training dataset and validation dataset from an FE simulation to interpret indentation load–displacement curves. Muliana et al. [18] generated an ANN model to approximate the FE load–depth curve and geometric parameters of heavy materials. Mahmoudi and Nourbakhsh [19] determined the three parameters of LUDWIG’s equation using the spherical indentation test and neural networks. Jeong et al. [6] proposed an FE-ANN model that integrates spherical indentation and ANN based on FE simulations to predict the uniaxial tensile flow of isotropic materials. Lu et al. [20] used several multi-fidelity ANN approaches to solve the inverse indentation problem, combining experimental data with simulation data and evaluating actual experiment data for several different metal materials. They suggested the indentation response from instrument indentation as the input parameter for predicting the stress–strain curve. However, so far, the research on predicting material anisotropy by neural network using indentation experiments has not been developed. Therefore, it would be valuable to study the anisotropy properties with neural networks instead of complex nonlinear relationships.

Compared with previous studies, this paper newly utilizes artificial neural network (ANN) techniques, which escapes the traditional complex dimensionless analysis to propose two prediction models for predicting anisotropy properties using a spherical indentation test applied to two different conditions. The proposed prediction models can not only predict the yield strength anisotropy of the sheet metal but also predict the Lankford coefficient that represents the anisotropy of the strain aspect in all orientations. The input parameters of the two models are proposed, which are the load–depth curve of the indentation response and the residual indentation mark, including indentation height and horizontal length. The stress–strain flow curve of the rolling direction, which is easier to obtain, is added as the input parameter of the first ANN prediction model for predicting yield strength ratio between any two orientations of the sheet metal, also evaluating the Lankford coefficient. The second model is the improved model, which predicts Young’s modulus of the material, the yield strength in one direction, and the hardness exponent. A new FE spherical indentation model is utilized for constructing the ANN prediction model that takes into account the effect of the compliance of the applied machine/frame/mounting material on the indentation response in the actual indentation test. We selected 1000 cases within the performance range of pure and alloyed metal materials and performed spherical indentation simulations through an FE model to obtain the training dataset. The characteristics of the load–depth curves and residual indentation marks obtained from the numerical simulation are extracted for use in the dataset of the ANN model. The performance of the proposed ANN prediction model was evaluated by comparison with obtained stress–strain curves and Lankford coefficient (r-value) in multiple loading directions using the uniaxial tensile test for different alloys, which is tested by actual spherical indentation tests. Moreover, to accurately measure the indentation profile obtained from the actual indentation tests, we used a machine vision method to determine each indentation mark’s central position automatically.

## 2. Materials and Methods

In order to construct an artificial neural network model that predicts the anisotropic properties of materials based on the indentation test, a dataset that is able to train the artificial neural network model with the physically related effective factors as the input parameters of the ANN is required. This paper constructed two kinds of ANN models for predicting plastic anisotropy properties, which is demonstrated in the flow chart in Figure 1.

The input parameters of the two models proposed included the load–depth curve and the residual indentation mark of the indentation response. A more readily available stress–strain flow curve in the rolling direction (RD) was added as an input parameter to the first ANN prediction model for predicting the yield strength ratio in any two orientations of the sheet metal while evaluating the Lankford coefficient. The second model was an improved model, which predicts Young’s modulus, the yield strength in one direction, and the hardness exponent in the condition of the material without knowing the properties of the RD orientation. A new FE spherical indentation model was utilized for constructing the training dataset of the ANN prediction models that take into account the effect of the compliance of the applied in the actual indentation test. Finally, the actual indentation test and uniaxial tensile test were utilized to verify the accuracy of the ANN model.

### 2.1. Mechanical Property Measurement of Materials

In order to verify the prediction accuracy of the two ANN prediction models composed of the datasets obtained from the FE indentation model for the actual experiment, this paper selected three different metal alloys with different elastoplastic properties as test materials to conduct indentation tests, steel (TRIP1180), aluminum alloy (AA6063-T6), and zinc alloy (Zn-Cu-Ti alloy). Since sheet metal forming is one of the most widely used processes in manufacturing, we selected the following thicknesses for the test materials: TRIP1180, 1.2 mm; AA6063-T6, 2 mm; and Zn-Cu-Ti, 2 mm. The uniaxial tensile test was used to measure the mechanical properties of these three materials. Moreover, the accuracy of the ANN model was verified by comparing the predicted value of the proposed ANN model with the target value of the uniaxial tensile test that generally obtains material anisotropy experiments.

Typical dog-bone-shaped tensile test specimens (ASTM-E8 standard) with a gauge width of 12.5 mm and a gauge length of 50 mm were fabricated at 15° intervals (0° to 90°) to the rolling direction. We tested only 0°, 45°, and 90° due to the anisotropic property of TRIP1180 not being large. Quasi-static uniaxial tensile tests based on displacement control were conducted using a strain rate of 0.003/s at 25 °C until a fracture occurred and the ARAMIS Digital Image Correlation system. Changes in the length and width of specimens during the tensile test were acquired from the recorded digital images to obtain the plastic strain of length ($\varepsilon_{l}{}^{p}$) and width ($\varepsilon_{w}{}^{p}$). The Lankford coefficient (r-value) is widely used as an indicator of sheet metal formability, characterizing its ability to resist thickening or thinning. The Lankford coefficient can be determined as shown in Equation (1), where the plastic strain along the specimen thickness ($\varepsilon_{t}{}^{p}$) can be estimated by the volume conservation principle using Equation (2). Figure 2 demonstrates the engineering stress–strain curves and r-value measured for these three materials. The yield strength and properties of the flow curves are listed in Table 1. The resulting material anisotropy properties were compared with the predictions of the proposed ANN model to compare the model fidelity.
(1)$$R=\frac{\varepsilon_{w}{}^{p}}{\varepsilon_{t}{}^{p}}$$
(2)$$\varepsilon_{t}{}^{p}=-\left(\varepsilon_{l}{}^{p}+\varepsilon_{w}{}^{p}\right)$$

An AIS2100 (Frontics Inc., Seoul, Korea), as shown in Figure 3, was used for the spherical indentation test with a force resolution of 5.6 *g*f and a displacement resolution of 0.1 μm. Because the rectangular 20 mm × 20 mm specimens for the indentation test were relatively thin, mounting material was prepared for the test. Moreover, the specimen surfaces needed to be finely polished and smooth so that no inclination on the surface of the specimen could affect the experimental process. Four repeated experiments were performed for each material using a tungsten carbide spherical indenter with a radius of 250 μm, and the maximum indentation depth in each experiment was 70 μm (test specimen thickness should be at least ten times the indentation depth or three times the indentation diameter according to ISO 14577-1). When the indenter reached the maximum depth, the dwell time was 500 ms. Figure 4 shows the load–depth curves calculated from repeated experiments for each material. TRIP1180 had a higher load for the 70 μm depth than the other two materials.

In order to accurately measure the residual indentation test when the indenter completely leaves the specimens, the resolution of the measuring device should be less than 1% of the maximum depth according to the macro-instrument indentation test measuring device standard (ISO14577-2). For the conditions of our actual indentation test, we utilized a 3D profile measurement device called ContourGT-K to extract characteristics, with a maximum vertical resolution of 0.01 nm. However, deviations in the extraction of indentation profiles in different orientations were due to the inability to accurately measure the center point of the residual indentation mark manually. This paper adopted the vision system to solve this difficulty, as depicted in Figure 5. The type of RGB image from the ContourGT-K is PNG. First, the RGB images were converted into HSV images because the hue values of HSV-type images can use colors to represent the depth of indentation more accurately than RGB images. Then the actual X, Y, and Z dimension size range provided by the ContourGT-K (Bruker Inc., Billerica, MA, USA) was set, which helped us accurately extract the indentation marks’ profiles. To accurately determine the material in any direction, the essential step was determining the center point of the indentation mark. The 0° yellow line that passes through the deepest point should be chosen as the baseline for extracting the X and Y axis pixels for the other direction lines. The final step was using the X and Y axis pixels to obtain the Z depth values and define the indentation profile. Table 2 provides the measurement results from the machine vision analysis. Compared with the traditional method of using software to determine the direction manually, machine vision can make more accurate measurements by automatically determining the center point of the residual indentation mark.

### 2.2. Input Parameters for the Proposed ANN Model

For predicting the anisotropy properties of various materials with the spherical indentation test, it is necessary to take into consideration the factors that affect the indentation response due to the properties of the material. Therefore, the selection of effective physically related influencing factors is crucial.

From the previous research [5,6,7,11,12,13,14,15], it was clear that the indentation response that affects the material properties mainly comes from two representative parts: the relationship between the indentation load and penetration depth (load–depth curve) and the geometric characteristics of the residual indentation marks. For the load–depth curve of the indentation test, various dimensionless functions using the ∏ theorem have already been proposed to predict the elastoplastic properties for an isotropic case [21]. The characteristics which are shown in Figure 6, the plastic work ratio ($\frac{W_{p}}{W_{t}}$), initial unloading slope (${\left.\frac{dP_{u}}{dh}\right|}_{h_{m}}$), loading curvature (*C*), and final indentation depth ($h_{r}$), were extracted to use the ∏ theorem functions to reverse analyze the elastoplastic properties of the material: Young‘s modulus (*E*), hardness exponent (*n*), and yield strength ($\sigma_{y}$). Dao, M. et al. [21] proposed that the plastic work ratio ($\frac{W_{p}}{W_{t}}$) and initial unloading slope (${\left.\frac{dP_{u}}{dh}\right|}_{h_{m}}$) can be utilized through the ∏_4_ and ∏_6_ theorems for solving Young‘s modulus (*E*) and the true projected contact area ($A_{m}$) of indenter. The dimensionless function ∏_1_ can be used for solving the stress of 0.033 strain ($\sigma_{0.033}$). Dao, M. et al. used the previously determined *E* and $\sigma_{0.033}$ to obtain the hardness exponent (n) through the ∏_2_ and ∏_3_ theorems. Finally, the yield strength ($\sigma_{y}$) was determined by the estimated properties. The curvature of the loading part for elastic–plastic materials of the spherical indentation test utilized in this paper was almost linear [22]. Therefore, the elastoplastic properties of the whole specimen were predicted by these influencing factors of the load–depth curve.

For the residual indentation mark, the difference according to the orientations of the material is highly dependent on the yield strength ratio (*m*) of the material in different orientations [13]. Furthermore, the yield strength ratio (*m*) is associated with the Lankford coefficient (*r*), which will be introduced in Section 2.3. We determined the residual indentation mark as the deformation normal to the specimen surface near the contact boundary under the fully unloaded condition. In Figure 7, if the deformed height (*h_z_*) of the original surface after removing the load was positive, it was piling up, and if it was negative, it was sinking in. The total height (*h_c_*) of the indentation mark was defined as the sum of the residual indentation depth (*h_f_*) and deformed height (*h_z_*), for example, the total height of rolling direction ($h_{cx}$) and transverse direction ($h_{cy}$). Yonezu et al. [13] applied the Π theorem to a dimensionless analysis; $h_{cx}$/$h_{cy}$ can be expressed as the following dimensionless function using Equation (3). The Π_6_ function shows the residual indentation height depends on the anisotropy properties, $\sigma_{r}$/$E^{*}$ and m. Yonezu et al. [13] used a large amount of FE data to summarize the relationship between $h_{cx}$/$h_{cy}$ and $\sigma_{r}$/$E^{*}$ by fitting the curve, showing the strong dependency of the yield strength ratio (*m*), as shown in Equation (4).
(3)$$\frac{h_{cx}}{h_{cy}}=\frac{h_{cx}/R}{h_{cy}/R}={\Pi \;}_{5}(\frac{\sigma_{r}}{E^{*}},\;n,\;m,\frac{h}{R})\;\approx \;{\Pi \;}_{6}(\frac{\sigma_{r}}{E^{*}},m)$$
(4)$$\frac{h_{cx}}{h_{cy}}=C_{1}(\frac{\sigma_{r}}{E^{*}})\;C_{2}$$
(5)$$\begin{array}{l} C_{1}=\;\left(m-1\right)\left(-0.0408m+0.2256\right)+1\\ C_{2}=\;\left(m-1\right)\left(-0.0521m+0.1885\right) \end{array}$$
where *C_1_* and *C_2_* are related to the material yield strength ratio (*m*), showing the coefficients *C_1_* and *C_2_* increase with the yield strength ratio (*m*), as shown in Equation (5). In the indentation mark, the radial length corresponding to the indentation height (*h_c_*) is defined as the residual indentation length (*L*). It decreases as the yield strength increases when the elastic modulus and indenter diameter are held constant [23] and can also be used as the influencing factor.

A review of the influencing factors of the spherical indentation test can reflect material properties. Figure 8 demonstrates the physically related influencing factors of the indentation response.

### 2.3. Dataset Acquisition Using FE Model for ANN Model

For obtaining the extensive training datasets for the ANN model, which was applied to predict the anisotropic properties of various metallic materials, this paper newly constructed an FE spherical indentation model, which is suitable for sheet metal by using the ABAQUS/standard. The indenter was assumed to be an analytically rigid model with a radius of 250 μm, and the specimen was defined as a deformable FE (C3D8R) model of 2 mm × 2 mm, with the size large enough to avoid the influence of outer boundary effects, and the height was the same as the actual specimens. The contact friction coefficient between the specimen and indenter surfaces was 0.12. Refined meshes were applied in the local contact region, as shown in Figure 9, in which the length of one side was 0.25 mm. A mesh size of 0.00625 mm and a 64,000 C3D8R element were applied to prevent the effect of the mesh size. To reduce computation time, we constructed a relatively coarse mesh in the area far from the contact surface and used a quarter-symmetric model. There were two processes of indenter movement, loading, and unloading, which were performed in the actual test. The indenter moved 70 μm from the specimen surface in the negative z-direction during the loading process, and for the unloading process, the indenter moved in the positive z-direction until it left the specimen. Figure 9 demonstrates the finite element model for this analysis process.

The linear elastic property was used, and the plasticity behaviors after the elastic property were defined, using the power-law strain hardening equation shown in Equation (6):(6)$$\begin{array}{c} \sigma =E\varepsilon ,\;for\;\sigma \leq \;\sigma_{y}\\ \sigma =R\varepsilon^{n},\;for\;\sigma \;\geq \;\sigma_{y} \end{array}$$

Anisotropic materials’ plasticity behaviors can be described using Hill’s yield criterion [24], which is one of the simplest and most widely used yield functions, given in Equation (7).
(7)$$f\left(\sigma \right)\;=\sqrt{F{\left(\sigma_{22}-\sigma_{33}\right)}^{2}+G{\left(\sigma_{33}-\sigma_{11}\right)}^{2}+H{\left(\sigma_{11}-\sigma_{22}\right)}^{2}+2L\tau_{23}^{2}+2M\tau_{31}^{2}+2N\tau_{12}^{2}}$$
where the *F*, *G*, *H*, *L*, *M,* and *N* are the anisotropic parameters [24] and were determined using Equation (8).
(8)$$\begin{array}{c} F=\frac{1}{2}\;\left(\frac{1}{R_{22}^{2}}+\frac{1}{R_{33}^{2}}-\frac{1}{R_{11}^{2}}\right);\;G=\;\frac{1}{2}\;\left(\frac{1}{R_{33}^{2}}+\frac{1}{R_{11}^{2}}-\frac{1}{R_{22}^{2}}\right)\\ H=\frac{1}{2}\;\left(\frac{1}{R_{11}^{2}}+\frac{1}{R_{22}^{2}}-\frac{1}{R_{33}^{2}}\right);\;L=\frac{3}{2R_{23}^{2}};\;M=\;\frac{3}{2R_{13}^{2}};\;N=\frac{3}{2R_{12}^{2}} \end{array}$$

In the ABAQUS/standard material model, Hill’s yield and its six parameters were used to define the plastic properties [15,25,26]. Among those parameters, *R*_11_, *R*_22_, and *R*_33_ were parameters for the normal direction, and *R*_12_, *R*_13_, and *R*_23_ were parameters for the shear direction. These parameters were used as properties representing anisotropy, with *R*_11_ defining the ratio (*m*) of the yield strength between the RD direction and the TD direction. The other Hill ratios were assumed to be 1. The anisotropy properties in the RD and TD directions, as well as properties for the other directions, can be derived using the characteristics of the residual indentation marks from the FE model as the basis for the ANN model. Moreover, *R*_11_ and *R*_33_, which relate to Hill’s yield criterion, are also associated with Lankford coefficients, as given in Equation (9), which means that the yield strength ratio (*m*) is related to the Lankford coefficient (r). Therefore, the relationship between the yield strength ratio and r-value ratio from RD to TD can be obtained through these two equations to predict the yield strength ratio (m) in the RD direction and any other direction. As above, the r-value ratio (*rr*) of the predicted direction to RD can be obtained by Equation (10). Nevertheless, to predict the r-value more accurately, we fitted several sets of data of the three actual materials and obtained a better relationship formula between the r-value ratio and yield strength ratio, as shown in Equation (11). The proposed fitting formula was more appropriate to the actual experimental data than the previous formula, as shown in Figure 10a, and the R-square of the proposed formula was 0.96088. The standard deviation of experiments was less than 0.1. The accuracy of the proposed formula was then verified with other experimental data for these three materials in Figure 10b, which showed a good agreement with experimental data. And Figure 11 shows that our actual experimental data were basically within the 95% confidence interval and prediction regions of the proposed formula. Thus, our prediction model can predict the r-value in different orientations by using the r-value at RD from the tensile test.
(9)$$R_{11}=\sqrt{\frac{r_{0}\left(r_{90}+1\right)}{r_{90}\left(r_{0}+1\right)}}=m,\;\;R_{33}=\sqrt{\frac{r_{0}\left(r_{90}+1\right)}{r_{90}+r_{0}}}=1$$
(10)$$rr=\frac{1}{\left(2m^{2}-1\right)}$$
(11)$$rr=\frac{0.99843}{m^{6.25156}}$$

To verify the constructed FE model, the anisotropic properties of TRIP1180 were applied to simulations, and the results were compared with the actual indentation test results. We confirmed that neither the loading nor unloading parts matched, as shown in Figure 12a. Since the properties of the load–depth curve are closely related to the stress–strain flow curve of the material, deviations of the finite element analysis from the experimental results can lead to deviations in the predictions of the material properties. It was thus important to modify our indentation FE model to match the actual experiment to ensure the accuracy of our FE-ANN model prediction. Therefore, to implement an actual experiment in the same analytical model, we considered the factors that influenced the experiment, particularly the elastic deformation (i.e., compliance) of the test device and the mounting material, proposed by Doerner and Nixis [27]. The total compliance (*C_total_*) of the actual experiment was the sum of specimen compliance (*C_specimen_*), device compliance (*C_device_*), and mounting material compliance (*C_mounting_*) [27,28,29], as shown in Figure 13.

Compliance (*C_device_* and *C_mounting_*) for the device and mounting material can be obtained from the difference between total compliance (*C_total_*) and specimen compliance (*C_specimen_*), as shown in Equation (12).
(12)$$C_{device}+C_{mounting}=C_{total}-C_{specimen}$$

The total compliance can be found as the inverse of the tangential slope ($\frac{dh}{dP}$) of the unloading curve at the maximum load point in the test. Next, we calculated the specimen compliance based on Sneddon’s elastic punch solution, which relates the reduced Young’s modulus of the specimen to the projected contact area (*A*) and is expressed by Equation (7), where *E_ind_* and *v_ind_* are the elastic modulus and Poisson’s ratio of the indenter [30], respectively, and *E_mat_* and *v_mat_* are the elastic properties and Poisson’s ratio of the indented TRIP1180 (*1.2 t*), which is tested by uniaxial tensile test, as are listed in Table 3.
(13)$$\begin{array}{c} C_{specimen}=\frac{1}{2E_{r}}\sqrt{\frac{\pi }{A}}\\ E_{r}={\left(\frac{1-v_{ind}{}^{2}}{E_{ind}}+\frac{1-v_{mat}{}^{2}}{E_{mat}}\right)}^{-1} \end{array}$$

A linear regression of *C_total_* and $1/\sqrt{A}$, which is shown in Equation (14), calculated m and b through a set of experiments with different maximum loads (*P_max_*, *h_max_*). The sum of the device and mounting compliance can be calculated after calculating the total compliance and specimen compliance.
(14)$$C_{total}=m\sqrt{\frac{1}{A}}+b$$

The spring with rigidity was obtained using Equation (15) as the reciprocal of the sum of device compliance and mounting compliance, which locates the bottom of the specimen in the FE model. Figure 14 demonstrates the FE model added a spring that represents the compliance of the frame of the test machine nodes on the baseline and connects it to the one spring node using kinematic coupling [31]. As shown in Figure 15, the stiffness of the spring was set to be symmetrical with respect to the zero point in the loading and unloading processes of the indentation test. The simulated result of the load–depth curve using the FE model with the added spring is shown in Figure 12a. Compared with the previous analysis, the deviation of maximum load, elastic area, plastic area, plastic area ratio, initial unloading slope, and final depth improved by 3.13%, 49.45%, 10.42%, 12.28%, 154.51%, and 32.54%, respectively. Moreover, we compared the experimental load–depth curve of AA6063-T6 and Zn-Cu-Ti with the FE simulation load–depth curve to verify whether adding the spring to the FE model applies to other materials, as shown in Figure 12b,c.
(15)$$Spring\;stiffness\left(K\right)\;=\frac{1}{C_{device}+C_{mounting}}$$

### 2.4. Anisotropy Artificial Neural Network Model

The two kinds of ANN models were constructed for predicting plastic anisotropy properties of sheet metal with respect to the yield strength and Lankford coefficient, which were trained with 1000 datasets. The first model exploited the RD properties as the input data to explore that the indentation test can make good predictions of yield strength ratios for anisotropic properties, which was because the properties of the RD orientation are easily obtained. This prediction model predicts the yield strength ratio (*m*) between any other direction and the RD to obtain the stress–strain curve for the other direction with reference to RD properties. Moreover, the Lankford coefficients for other directions were evaluated based on the previously proposed relationship between the value ratio of RD and the yield strength ratio in the predicted direction. The ANN structure includes multiple inputs that are multiplied by different weights and then uses a mathematical function to determine whether it can stimulate neurons, with an activation function for calculating the output of the artificial neuron. This ANN belonged to the multi-layer perceptron class, in which five hidden layers are set in the input and output layers. The backward propagation algorithm was used to continuously adjust the weights and thereby minimize the error between the output values, and the correct answer was used for training the ANN. The input layer of the first model consisted of 18 neurons, as shown in Table 4, the output layer had 1 neuron, representing the yield strength ratio (*m*). In the supervised training of the ANN, the following hyper-parameters were used: number of epochs: 10,000; batch size: 64; and choice of activation function: sigmoid. In addition, in the training process of this research, the initial learning rate was set to 0.005. The mean absolute error (*MAE*) was set as the loss function, as defined in Equation (16):(16)$$MAE=\frac{1}{n}\sum_{i=1}^{n}\left|y_{target}-x_{pred.}\right|$$
where $y_{target}$ and $x_{pred.}$ indicate the target value and prediction value, n is the total number of the dataset. In addition, the elastic and plastic properties of a material can also be predicted using the indentation test. Therefore, we took not only the yield strength ratio (*m*) but also the elastic modulus (*E*), yield strength ($\sigma_{y}$) in any orientation, and hardness exponent (*n*) as prediction targets and constructed an improved model from our previous model. The model did not employ the RD material properties as the input parameters but directly used the characteristics of the sheet metal’s indentation response with the input layer to 15 neurons and the output layer to 4 neurons.

For 1000 training datasets, one thousand elastoplastic parameter conditions commonly found in pure and alloyed engineering metals [20] were set as the FE model’s material properties to construct a large simulation, as shown in Table 5. The load–depth corresponding to each set of parameters and the profiles of all the residual indentation marks were extracted for constituting the dataset with the predicted material properties. For preventing the overfitting problem, one thousand datasets were divided into training datasets and test datasets with no overlap, at the ratio of 0.8:0.2. Figure 16 describes the procedure for constructing the datasets and training the ANN model.

## 3. Results

Two kinds of ANN prediction models proposed were applied for predicting the anisotropy properties of five materials, with the one test closest to the result obtained from the corresponding FE indentation model selected in five repeated indentation tests. To evaluate the quality of the ANN, we used correlations (R-squared) that represented the statistical relationship between the target and predicted variables. Figure 17 and Table 6 show the correlation result of the validation dataset, which was not included in the training dataset of the prediction models, in the first prediction model and second prediction model. The correlation of the first model was higher than 0.997, and the second model was higher than 0.989, which indicates that the proposed prediction models have an excellent ability to predict the target value.

The first prediction model was verified by predicting the material properties of different orientations of the TRIP1180, AA6063-T6, and Zn-Cu-Ti alloy sheets. We compared the prediction results with the uniaxial tensile test flow curves to evaluate the prediction model’s performance. Figure 18 and Table 7 show the stress–strain flow curve predicted by the first model for the three materials (symbol line) and the curves from the uniaxial tensile testing (solid line). The deviation between the predicted flow curve area and the actual experimental flow curve area was used to judge the quality of the prediction result. For TRIP1180, which had higher strength than the other test materials, the deviation between the 45° and 90° flow curve prediction results and the actual experimental result was less than 1%. However, even for the AL6063-T6 and Zn-Cu-Ti alloy materials, which had larger anisotropy and lower strength than the TRIP1180 steel, the biggest deviation in the flow curve area was not more than 3.4%. In addition, the Lankford coefficient was estimated through the prediction model, as shown in Figure 19. It is clear that the predictions and experimental results showed consistent trends and extremely high accuracy.

Figure 20 depicts the flow curves predicted by the second ANN prediction model, showing the 45° and 90° results for TRIP1180 and 15° intervals from 0° to 90° for the AL6063-T6 and Zn-Cu-Ti alloy based on the 0° direction, the specific predicted data are shown in Table 8. The predicted flow curve deviations for the TRIP1180, AA6063-T6, and Zn-Cu-Ti alloys using this prediction model were less than 5%. Therefore, regardless of the magnitude of the yield strength, hardness exponent, or yield strength ratio of these three materials, the predicted results were in good agreement with the actual experimental results. Furthermore, although the predicted Young’s modulus values for the AA6063-T6 and Zn-Cu-Ti alloy, which have relatively small Young’s modulus values, had sizeable numerical deviations, the prediction error for TRIP1180, which had a relatively large Young’s modulus, was less than 1%.

Moreover, we compared the results of yield strength ratio and Lankford coefficient of the two ANN prediction models, which are depicted in Figure 21 and Figure 22. The figures clearly show that the prediction accuracy of the first prediction model was higher than that of the second model due to the additional RD properties as input parameters and a small number of prediction targets with the less difficult for prediction. However, the second model comprehensively showed that the properties of the material, including the elastoplastic properties and anisotropy properties of the material, can be estimated through the indentation test. Therefore, when the experimental data in the RD direction can be obtained, the first prediction model can be used to accurately predict the yield strength ratio and the Lankford coefficient of the material. If there is no condition to obtain the attributes in the RD direction, the second prediction model can be used for comprehensive attribute prediction.

## 4. Conclusions

This paper proposed two kinds of ANN models for predicting the anisotropy properties with different materials based on the load–depth curve and residual indentation mark derived from the spherical indentation test, which measures the mechanical anisotropy more simply than a tensile test. For obtaining a dataset with a large number with high efficiency and low cost for training the ANN prediction model, this paper newly constructed the FE spherical indentation model with the compliance, which was consistent with the actual indentation test to perform a large simulation with the material in a range of pure and alloyed metal materials. The compliance of the experimental device and mounting was determined by using the load–depth curve from actual experiments. Moreover, in order to verify the performance of the two proposed ANN models, we compared the ANN prediction results with the experiment results of the uniaxial tensile tests in TRIP1180, AA6063-T6, and Zn-Cu-Ti alloy. To gain reliable values about the indentation marks in the experiment, the machine vision system was adopted since this system can reduce the errors caused by manual operations. Under these circumstances, final conclusions can be drawn.
An ANN model for predicting anisotropic properties of materials was constructed, and this model can replace the conventional dimensionless analysis with complex procedures to derive the analysis function.The proposed two types of artificial neural network models can predict the anisotropic properties of materials, where the first prediction model, with RD characteristics as input parameters, can predict the yield strength ratio and the Lankford coefficient well. On the other hand, in cases without RD characteristics as input parameters, the second model provides a comprehensive prediction of the material’s properties, including its elastic–plastic and anisotropic properties.The predicted yield strength ratios using the first model of TRIP1180, AA6063-T6, and Zn-Cu-Ti alloy had a maximum deviation from the experimental results of 0.6%, 2.6%, and 4.9%, respectively. At the same time, the Lankford coefficient predicted by our model was consistent with the experimental results. The deviations in the predicted stress–strain flow curves were all less than 5%. Furthermore, the flow curve predicted for each material using the improved ANN model showed a maximum deviation from the tensile test of 4.7%.In future work, the deeper prediction ANN model for the anisotropy of Young’s modulus and hardness exponent can be constructed to comprehensively predict the anisotropy properties of materials with larger Young’s modulus and hardness exponent. Furthermore, it is possible to improve the ANN model, which can predict the anisotropy properties by utilizing the Chaboche model after undergoing the complex stress state.

The proposed method, which uses the spherical indentation test with the ANN model, showed excellent predictions on the anisotropy of yield strength and Lankford coefficient in different directions. This paper is a great advance in material property prediction. It also heralds that the data networks can be used widely for the prediction of various properties of materials.

## Figures and Tables

**Figure 1 materials-15-01714-f001:**
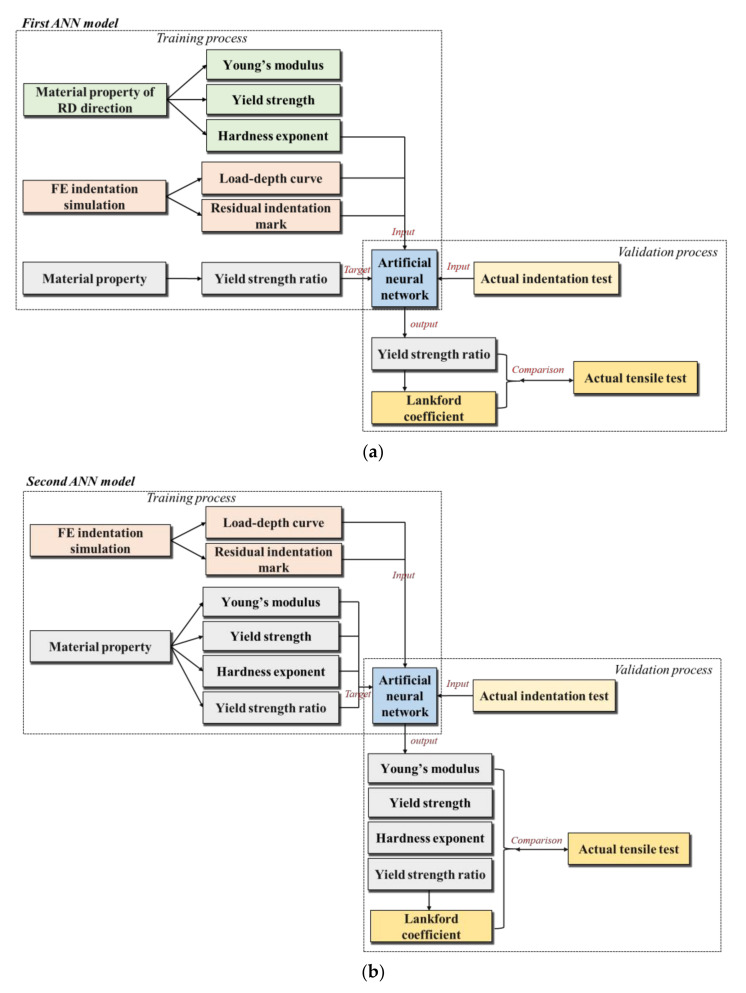
Flow chart for predicting anisotropy properties with two kinds of ANN model: (**a**) First ANN model, (**b**) Second ANN model.

**Figure 2 materials-15-01714-f002:**
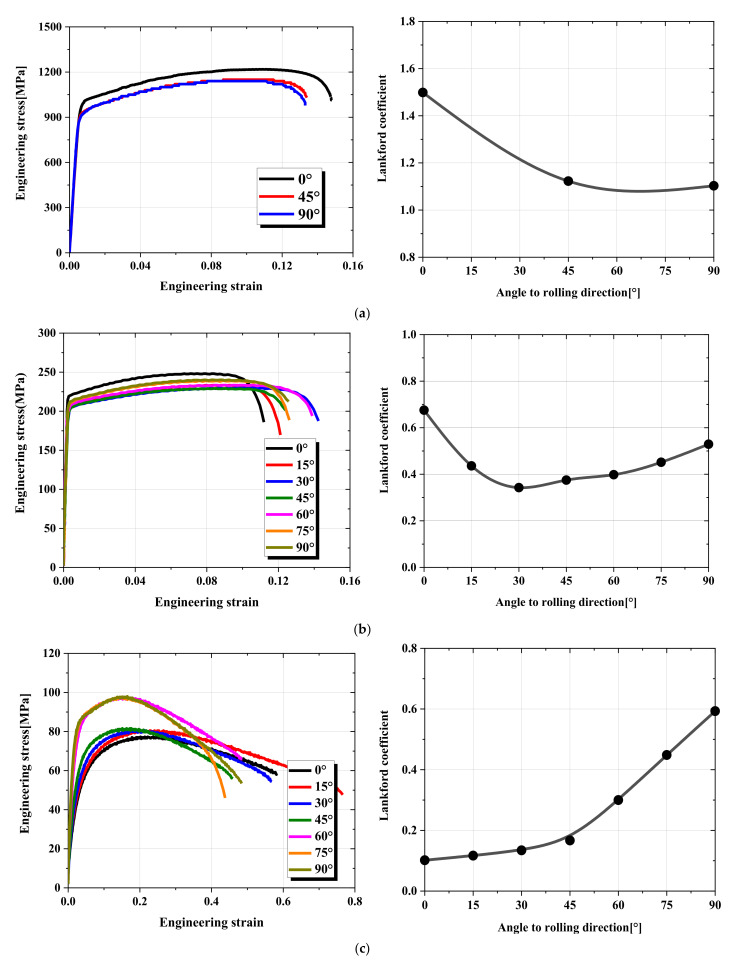
Engineering stress–strain curves and Lankford coefficient (r-value): (**a**) TRIP1180, (**b**) AA6063-T6, (**c**) Zn-Cu-Ti alloy.

**Figure 3 materials-15-01714-f003:**
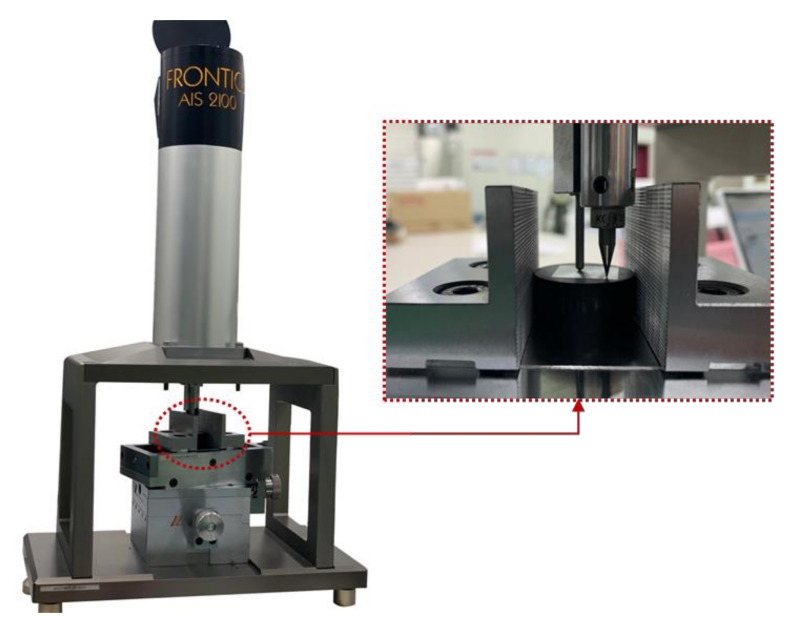
The AIS2100 indentation test device.

**Figure 4 materials-15-01714-f004:**
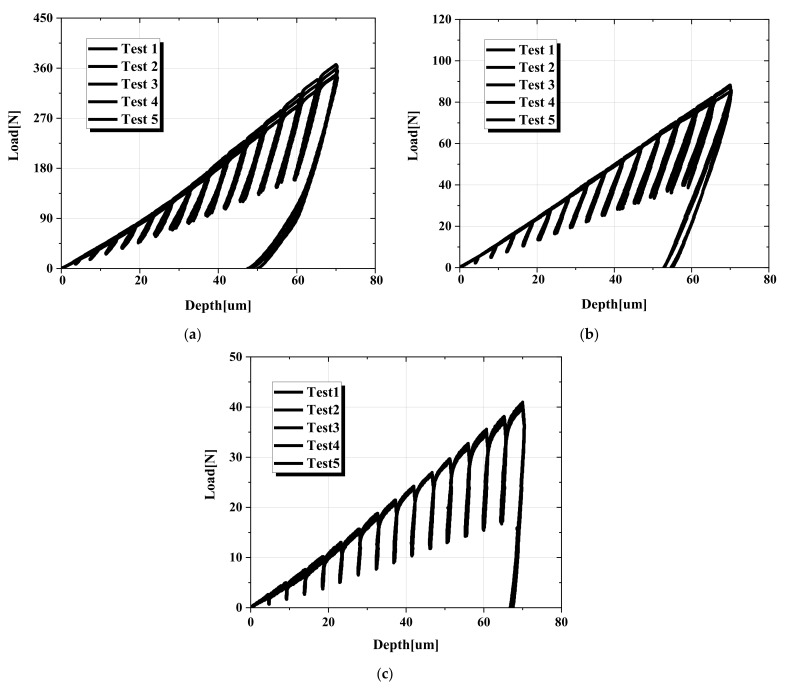
Load–depth curves from the indentation tests: (**a**) TRIP1180, (**b**) AA6063-T6, (**c**) Zn-Cu-Ti alloy.

**Figure 5 materials-15-01714-f005:**
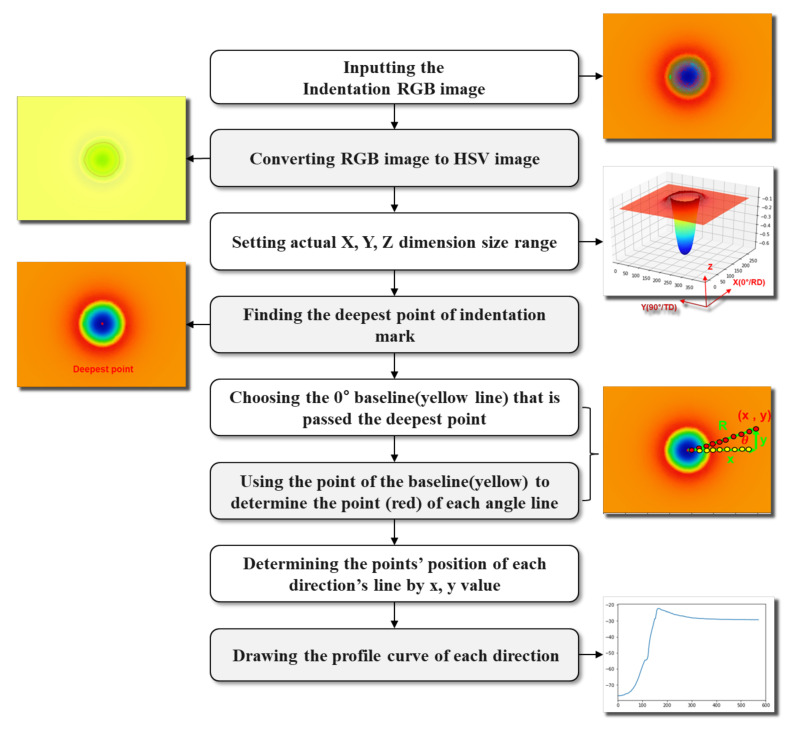
Schematic diagram of the residual indentation mark analysis procedure using machine vision.

**Figure 6 materials-15-01714-f006:**
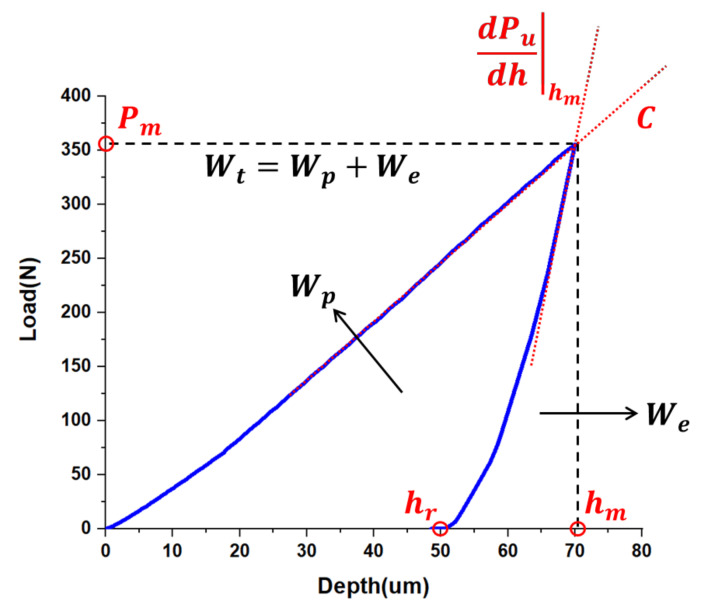
Load–depth curve of indentation test and its related factors.

**Figure 7 materials-15-01714-f007:**
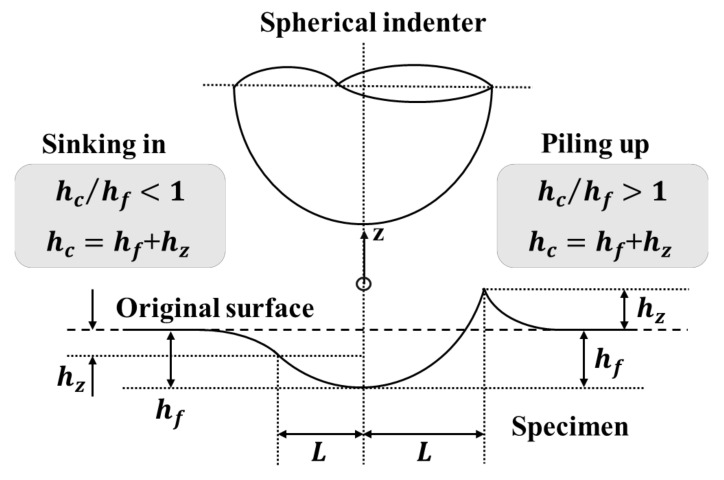
Schematic diagram of the impression morphology caused by spherical indentation.

**Figure 8 materials-15-01714-f008:**
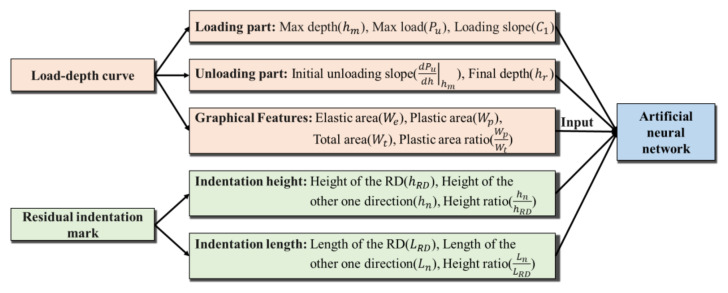
Schematic diagram of the influencing factors for the artificial neural network.

**Figure 9 materials-15-01714-f009:**
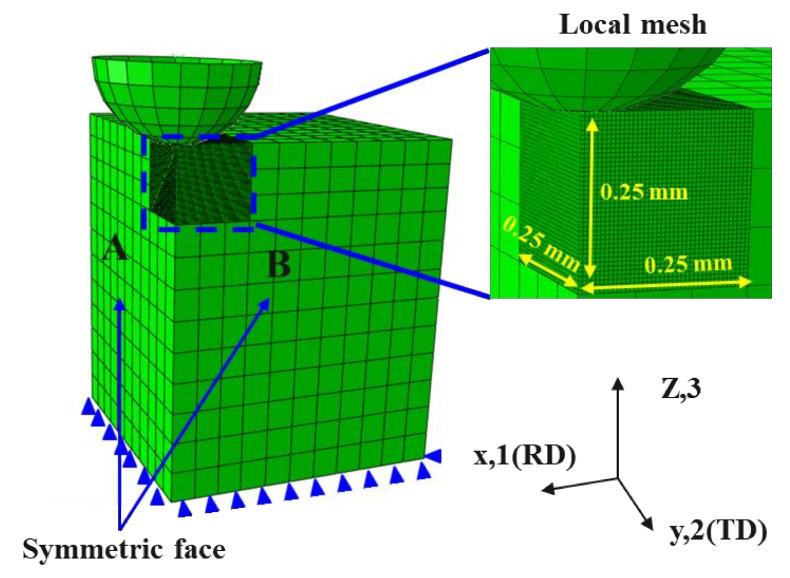
FE spherical indentation model and mesh settings.

**Figure 10 materials-15-01714-f010:**
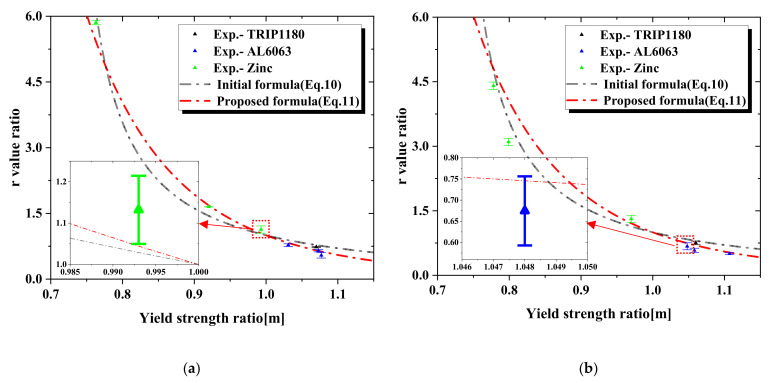
Formula for the r-value ratio and yield strength ratio: (**a**) Fitting process, (**b**) Validation process.

**Figure 11 materials-15-01714-f011:**
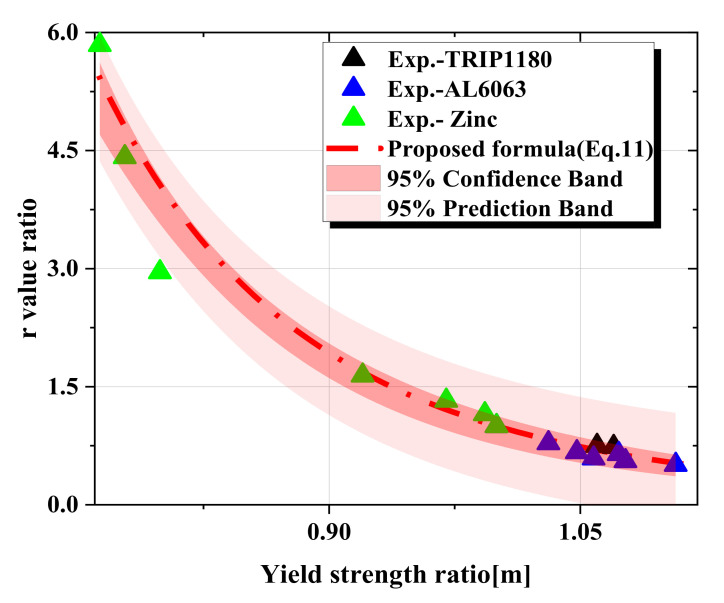
The confidence band and prediction band of the proposed formula.

**Figure 12 materials-15-01714-f012:**
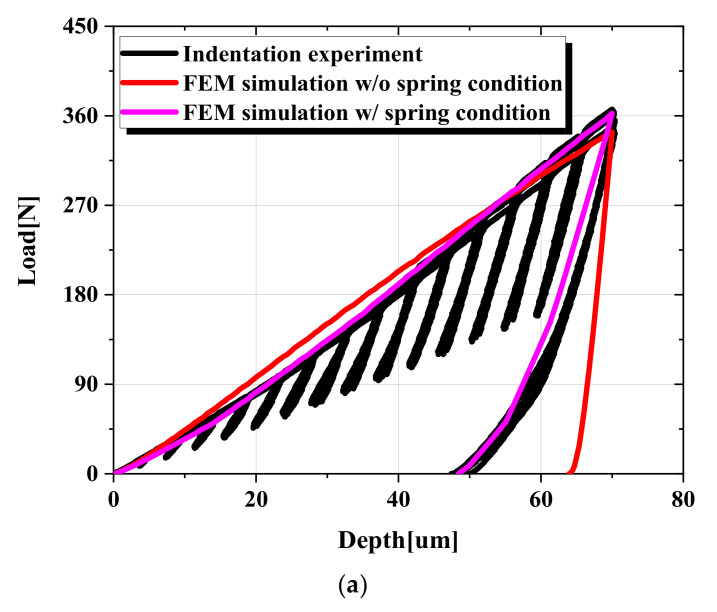
Comparison of results after considering compliance (Comparison of the load–depth curve): (**a**) TRIP1180, (**b**) AA6063-T6, (**c**) Zn-Cu-Ti alloy.

**Figure 13 materials-15-01714-f013:**
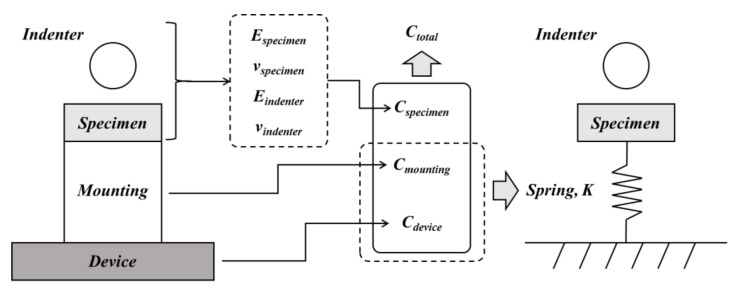
The schematic diagram for considering compliance.

**Figure 14 materials-15-01714-f014:**
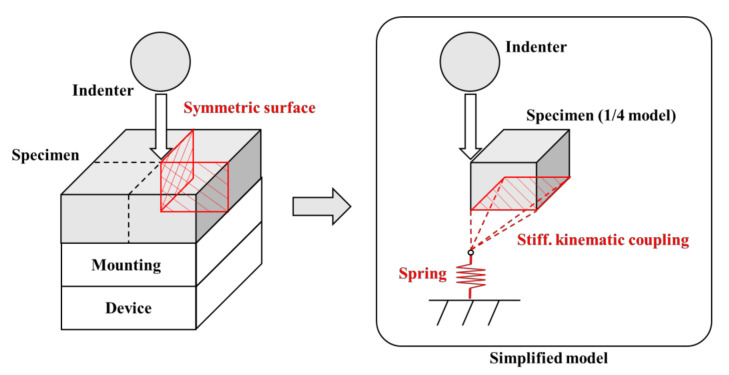
FE model with spring boundary condition.

**Figure 15 materials-15-01714-f015:**
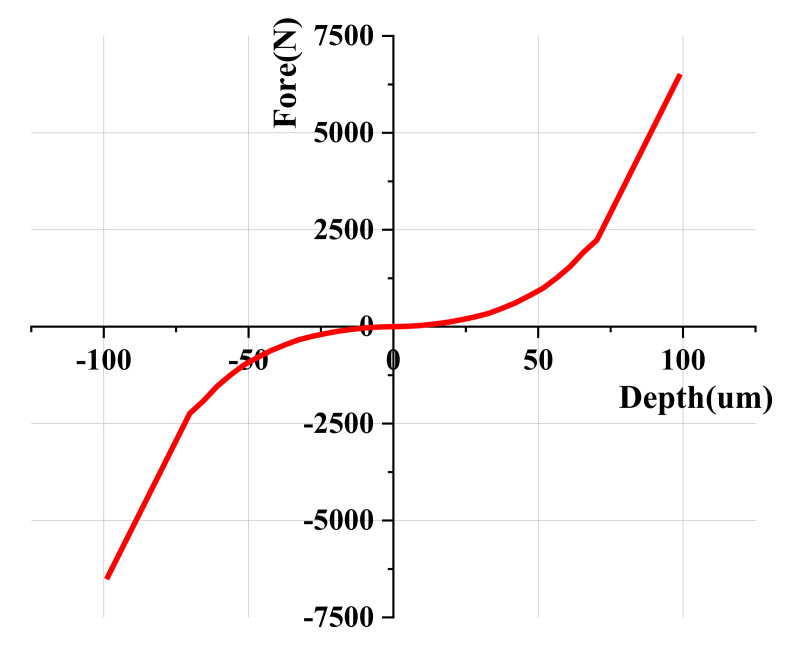
The characteristics of the spring boundary condition.

**Figure 16 materials-15-01714-f016:**
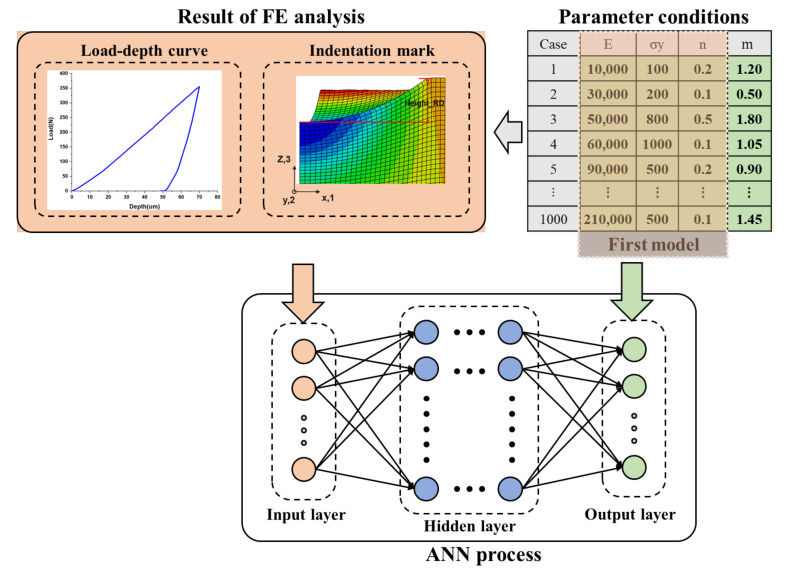
Flow chart for obtaining the training dataset for the ANN model.

**Figure 17 materials-15-01714-f017:**
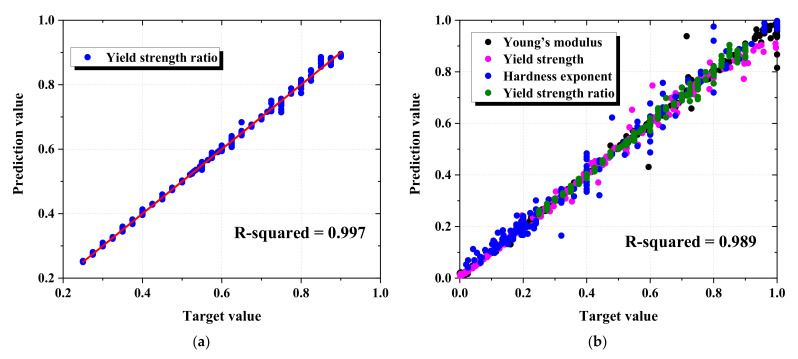
Correlation of two prediction models: (**a**) First model, (**b**) Second model.

**Figure 18 materials-15-01714-f018:**
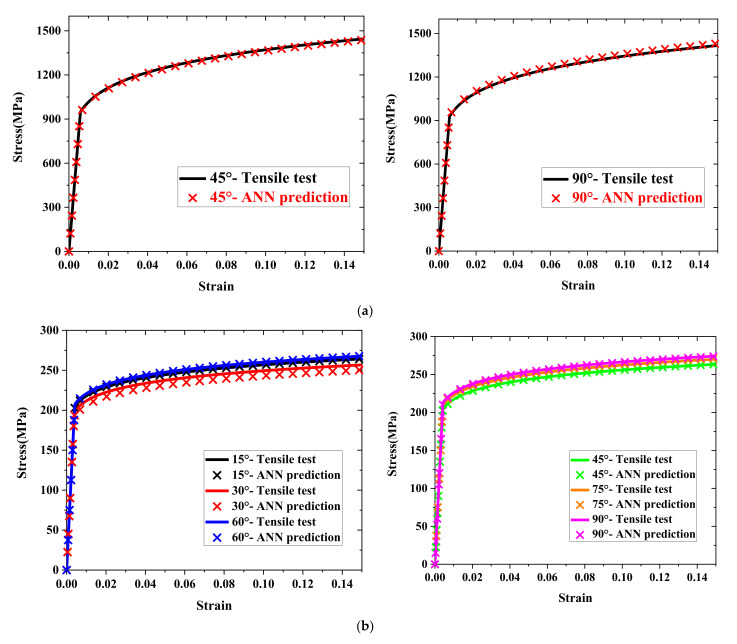
Stress–strain flow curves predicted by the first ANN model: (**a**) TRIP1180, (**b**) AA6063-T6, (**c**) Zn-Cu-Ti alloy.

**Figure 19 materials-15-01714-f019:**
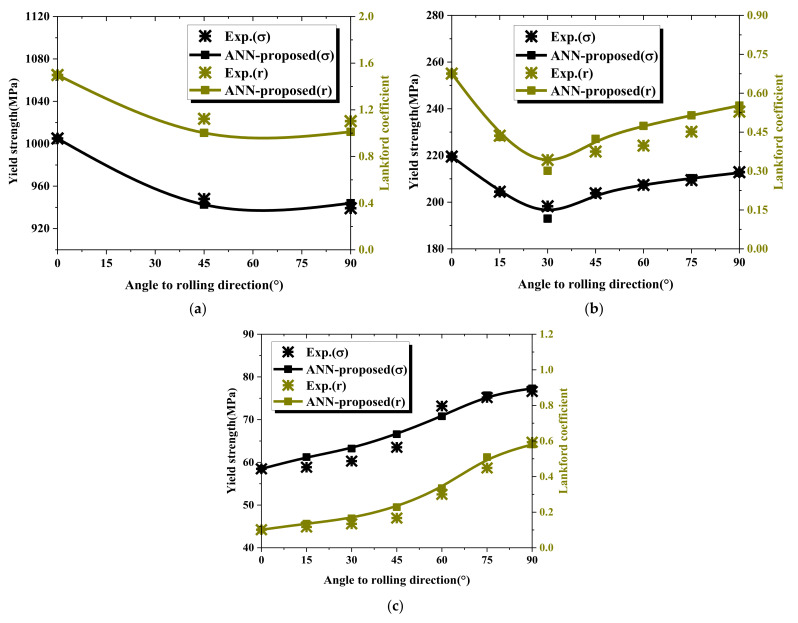
Lankford coefficient (r-value) predicted by the first ANN model: (**a**) TRIP1180, (**b**) AA6063-T6, (**c**) Zn-Cu-Ti alloy.

**Figure 20 materials-15-01714-f020:**
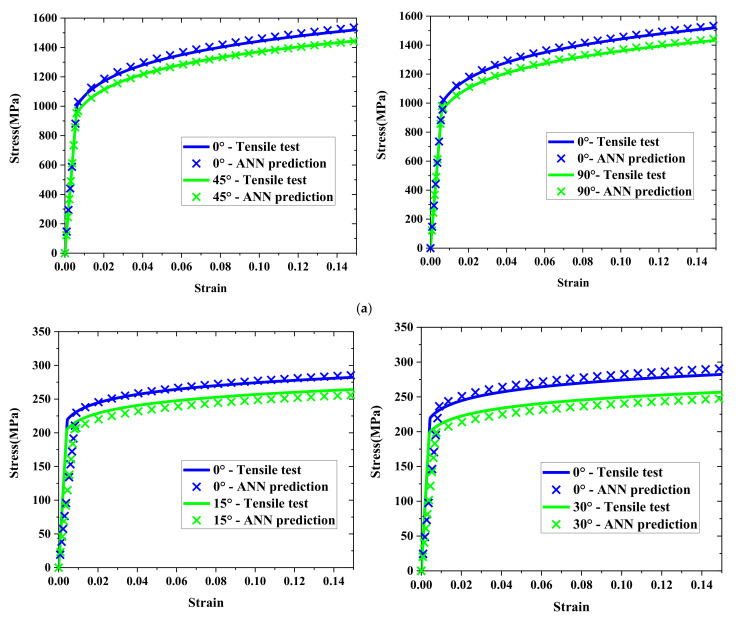
Stress–strain flow curves predicted by the second ANN model: (**a**) TRIP1180, (**b**) AA6063-T6, (**c**) Zn-Cu-Ti alloy.

**Figure 21 materials-15-01714-f021:**
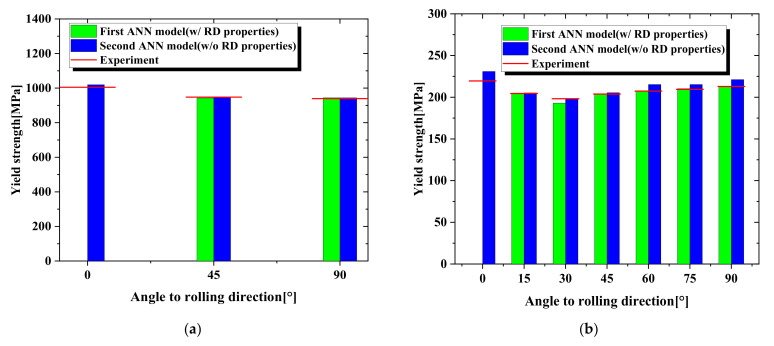
Comparison of yield strength by two prediction models: (**a**) TRIP1180, (**b**) AA6063-T6, (**c**) Zn-Cu-Ti alloy.

**Figure 22 materials-15-01714-f022:**
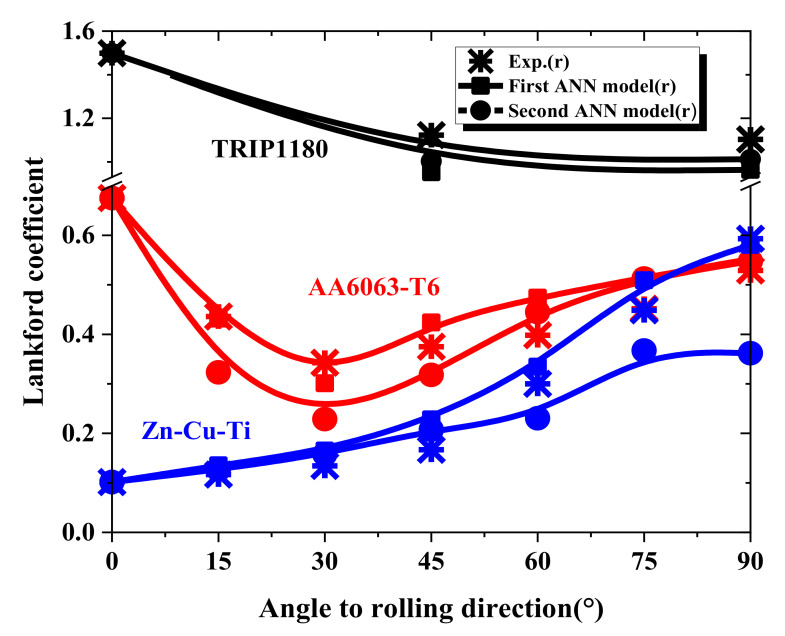
Comparison of Lankford coefficient by two prediction models for TRIP1180, AA6063-T6, and Zn-Cu-Ti alloy.

**Table 1 materials-15-01714-t001:** Engineering stress–strain data for each material.

		Young’s Modulus (MPa)	Yield Strength (MPa)	Strain Hardening Exponent (n)	Lankford Coefficient (r)
TRIP1180	0°	161,994	1005	0.13	1.499
	45°		948		1.123
	90°		939		1.103
AA6063-T6	0°	50,000	219	0.0712	0.675
	15°		204		0.436
	30°		198		0.343
	45°		203		0.375
	60°		207		0.398
	75°		209		0.452
	90°		212		0.529
Zn-Cu-Ti alloy	0°	127,700	58	0.306	0.101
	15°		59		0.117
	30°		60		0.134
	45°		63		0.167
	60°		73		0.300
	75°		75		0.449
	90°		76		0.593

**Table 2 materials-15-01714-t002:** Analysis results from using machine vision for TRIP1180, AA6063-T6, and Zn-Cu-Ti alloy.

	Direction	TRIP1180	AA6063-T6	Zn-Cu-Ti Alloy
Indentation height (μm)	0°	52.315	58.747	76.597
	15°	−	63.026	74.496
	30°	−	65.357	72.689
	45°	53.521	63.161	70.363
	60°	−	60.717	68.673
	75°	−	59.539	65.604
	90°	53.587	59.109	65.520
Indentation length (μm)	0°	158.385	160.625	191.154
	15°	−	170.201	190.768
	30°	−	175.103	187.953
	45°	159.306	170.885	184.407
	60°	−	165.869	178.858
	75°	−	162.791	178.190
	90°	160.481	162.335	177.520

**Table 3 materials-15-01714-t003:** Relevant properties of TRIP1180 and the indenter.

	Indenter	Material
E (MPa)	700,000	161,994
v	0.31	0.31

**Table 4 materials-15-01714-t004:** Relevant properties of TRIP1180 and the indenter.

Load–Depth Curve	Residual Indentation Mark	Properties of the RD Stress–Strain Curve (Only Works on the First Model)
Max depth, $h_{m}$ Max load, $P_{u}$ Elastic area, $W_{e}$ Plastic area, $W_{p}$ Total area, $W_{t}$ Plastic area ratio, $W_{p}/W_{t}$ Loading slope, $C_{1}$ Initial unloading slope, ${\left.\frac{dP_{u}}{dh}\right|}_{h_{m}}$ Final depth, $h_{r}$	Height of the RD, $h_{RD}$ Height of the other one direction, $h_{n}$ Height ratio, $h_{n}/h_{RD}$ Length of the RD, $L_{RD}$ Length of the other one direction, $L_{n}$ Length ratio, $L_{n}/L_{RD}$	Young’s modulus, E RD yield strength, $\sigma_{y\_RD}$ Hardness strength, n

**Table 5 materials-15-01714-t005:** Range of elastoplastic parameter conditions commonly found in pure and alloyed engineering materials.

Properties	Range
Young’s modulus (E)	5~210 GPa
Yield strength ($\sigma_{y}$)	30~3000 MPa
Hardness exponent (n)	0~0.5
Yield strength ratio (m)	0.1~2

**Table 6 materials-15-01714-t006:** Correlation (R-squared) and loss (MAE) of the proposed prediction models.

First Model	Yield Strength Ratio (m)			
Correlation (R-squared)	99.764%			
Loss (MAE)	0.285%			
**Second model**	**Young’s modulus (E)**	**Yield strength** ($\sigma_{y}$)	**Hardness exponent (n)**	**Yield strength ratio(m)**
Correlation (R-squared)	99.004%	99.079%	98.231%	99.400%
Loss (MAE)	1.161%	1.509%	2.506%	0.913%

**Table 7 materials-15-01714-t007:** Predicted properties of the stress–strain curve.

Direction		Yield Strength $\left(\sigma_{y}\right)$			
		Tensile Test Result (MPa)	ANN Prediction Result (MPa)	Deviation	Deviation in the Flow Curve Area
TRIP1180	45°	948.023	942.603	0.572%	0.499%
	90°	939.164	944.000	0.515%	0.433%
AA6063-T6	15°	204.497	204.404	0.042%	0.042%
	30°	198.227	192.956	2.578%	2.445%
	45°	203.815	203.807	0.004%	0.003%
	60°	207.376	207.478	0.050%	0.045%
	75°	209.392	210.239	0.408%	0.372%
	90°	212.856	212.612	0.117%	0.105%
Zn-Cu-Ti alloy	15°	58.859	61.238	4.042%	2.787%
	30°	60.291	63.221	4.860%	3.347%
	45°	63.528	66.586	4.814%	3.331%
	60°	73.177	70.79	3.262%	2.274%
	75°	75.169	75.707	0.716%	0.496%
	90°	76.602	77.306	0.919%	0.637%

**Table 8 materials-15-01714-t008:** Predicted properties of TRIP1180 (1.2 t), AA6063-T6, and Zn-Cu-Ti alloy.

Material	Properties	Direction	Tensile Test Result	ANN Prediction Result
TRIP1180	Young’s modulus (E)	0°–45°	161,994	163,075
		0°–90°	161,994	163,266
	Yield strength ($\sigma_{y}$)	0°–45°	1005/948.023	1019.760/948.614
		0°–90°	1005/939.164	1012.740/943.840
	Hardness exponent (n)	0°–45°	0.130	0.129
		0°–90°	0.130	0.130
	Yield strength ratio (m)	0°–45°	1.060	1.075
		0°–90°	1.070	1.073
	Deviation of the flow curve area	0°–45°	1.172%/0.045%	
		0°–90°	0.924%/0.695%	
AA6063-T6	Young’s modulus (E)	0°–15°	50,000	25,509
		0°–30°	50,000	27,065
		0°–45°	50,000	25,587
		0°–60°	50,000	23,739
		0°–75°	50,000	23,354
		0°–90°	50,000	23,193
	Yield strength ($\sigma_{y}$)	0°–15°	219.520 /204.497	230.893 /205.238
		0°–30°	219.520 /198.227	235.867 /198.374
		0°–45°	219.520 /203.815	231.663 /205.375
		0°–60°	219.520 /207.376	230.116 /215.263
		0°–75°	219.520 /209.392	227.903 /218.089
		0°–90°	219.520 /212.856	228.652 /221.134
	Hardness exponent (n)	0°–15°	0.0712	0.0755
		0°–30°	0.0712	0.0733
		0°–45°	0.0712	0.0744
		0°–60°	0.0712	0.0735
		0°–75°	0.0712	0.0784
		0°–90°	0.0712	0.077
	Yield strength ratio (m)	0°–15°	1.073469	1.125
		0°–30°	1.107423	1.189
		0°–45°	1.077061	1.128
		0°–60°	1.058566	1.069
		0°–75°	1.048374	1.045
		0°–90°	1.031311	1.034
	Deviation of the flow curve area	0°–15°	0.483%/4.504%	
		0°–30°	1.604%/4.629%	
		0°–45°	0.388%/4.361%	
		0°–60°	1.848%/2.667%	
		0°–75°	1.930%/1.622%	
		0°–90°	2.027%/2.214%	
Zn-Cu-Ti alloy	Young’s modulus (E)	0°–15°	127,700	185,697
		0°–30°	127,700	180,248
		0°–45°	127,700	170,498
		0°–60°	127,700	167,321
		0°–75°	127,700	138,878
		0°–90°	127,700	138,642
	Yield strength ($\sigma_{y}$)	0°–15°	58.477 /58.859	63.538 /65.843
		0°–30°	58.477 /60.291	61.618 /66.114
		0°–45°	58.477 /63.528	59.194 /66.361
		0°–60°	58.477 /73.177	58.143 /66.298
		0°–75°	58.477 /75.169	54.336 /66.752
		0°–90°	58.477 /76.602	54.514 /66.807
	Hardness exponent (n)	0°–15°	0.306	0.271
		0°–30°	0.306	0.278
		0°–45°	0.306	0.288
		0°–60°	0.306	0.297
		0°–75°	0.306	0.311
		0°–90°	0.306	0.311
	Yield strength ratio (m)	0°–15°	0.993	0.965
		0°–30°	0.969	0.932
		0°–45°	0.920	0.892
		0°–60°	0.799	0.877
		0°–75°	0.778	0.814
		0°–90°	0.763	0.816
	Deviation of the flow curve area	0°–15°	1.244%/0.867%	
		0°–30°	0.417%/2.576%	
		0°–45°	0.326%/2.760%	
		0°–60°	3.284%/3.049%	
		0°–75°	0.120%/3.285%	
		0°–90°	0.013%/4.597%	

## Data Availability

Not applicable.

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
