# Peer review of "Artificial Neural Networks for Predicting Plastic Anisotropy of Sheet Metals Based on Indentation Test"

_materials, 2022, doi:10.3390/ma15051714_

Round 1

Reviewer 1 Report

The study is very interesting, clearly described and well-structured.

The authors describe two artificial neural network models for predicting plastic anisotropy properties of sheet metal using spherical indentation test. A FE spherical indentation model was constructed in order to obtain a significant dataset. Experimental indentations have been carried out on three materials.

Below, the authors may find few specific suggestions/comments:

  • As indicated in the Instructions for Authors, the journal does not have strict formatting requirements, but all manuscripts must contain after the Introduction the following sections: Materials & Methods, Results, and Conclusions (only this latter is actually already included in the manuscript).
  • Introduction: the context of the study and the purpose have been well highligted. The state-of-art has been well structured and presented. The aim of the study is also clear, but the main conclusions should be better presented.
  • 4 – Line 161: “From the previous research,..” It seems that the authors refer to a specific work, therefore, a reference should be added.
  • Figure 17 and 19: The font of the captions are a little bit bigger than the others.
  • Table 3: Add a full stop at the end of the caption (just to use the same style of the others in the text).

Reviewer 2 Report

The paper “Artificial neural networks for predicting plastic anisotropy of sheet metals based on indentation test” addresses profile of the residual indentation, include the height and length in different orientations used to analyze the anisotropic properties of the material. It is of interest and novelty, so I suggest considering it for publication after minor changes:

  • A graphical abstract would add interest to catch the eye
  • Special emphasis should be placed on novelty in both the introduction and the abstract
  • The paper is written in a clear way, a short overview of the literature has been done. I miss some previous articles addressing material anisotropy in other materials:

https://doi.org/10.1016/j.jmatprotec.2021.117271

  • The sequence of three figures in a row without text in between becomes very strange in Figure 3-4-5.
  • Mesh size sensitivity analysis has been carried out for the definition of the optimal number of elements.
  • You can include more data on the number of elements in the finite element analysis.
  • Has data on the hyperparameters used in the neural network used.
  • Could you give in the conclusions some guidelines for future lines of action?

Overall, the article is well organized and correctly presented after small changes I find nothing against its publication.

Reviewer 3 Report

In my opinion the paper is well written and presents two proposals of ANN models for predicting the anisotropy properties with different materials based on the load-depth curve and residual indentation mark derived from the spherical indentation tes. Based on this, it can be concluded that using such simple test and idea of Authors, it is possible to obtain excellent conformity between experiemental data and predicted values. On the other hand several adjustmensts are required. Below, Please follow the points listed below. 1. Please explain the source of Eq. 1. If it is dimensionless, please interprete the constants like. 0.2256 etc. How about any statistical analysis of the model Eq. 1 2. FEM silumations – Can the Authors explain the main idea of the model in Eq. 4 for loading above yielding stress. It seems that is suitable in uniaxial loading, but how about complex stress state. Did the Authors investigate other models like Chaboche etc? 3. Table 1 – Please add source of such data and any experimental results including variations of tested values 4. Fig. 14 – How about scatter bands? Did the Authors calculate any statistical output? Please present it In general, I would like to express my positive opinion about paper.
